# Hemp Seeds (*Cannabis sativa* L.) as a Valuable Source of Natural Ingredients for Functional Foods—A Review

**DOI:** 10.3390/molecules29092097

**Published:** 2024-05-01

**Authors:** Virginia Tănase Apetroaei, Eugenia Mihaela Pricop, Daniela Ionela Istrati, Camelia Vizireanu

**Affiliations:** Faculty of Food Science and Engineering, Dunarea de Jos University of Galati, 111 Domneasca Street, 800201 Galati, Romania; virginia.ape7@gmail.com (V.T.A.); mihaela.pricop@ugal.ro (E.M.P.); camelia.vizireanu@ugal.ro (C.V.)

**Keywords:** *Cannabis sativa* L., nutritional components, bioactive compounds, antinutritional compounds, functional foods

## Abstract

Hemp (*Cannabis sativa* L.) has experienced a significant resurgence in popularity, and global interest in diversifying its use in various industries, including the food industry, is growing. Therefore, due to their exceptional nutritional value, hemp seeds have recently gained increasing interest as a valuable ingredient for obtaining high-quality foods and dietary supplements. Hemp seeds stand out for their remarkable content of quality proteins, including edestin and albumin, two distinct types of proteins that contribute to exceptional nutritional value. Hemp seeds are also rich in healthy lipids with a high content of polyunsaturated fatty acids, such as linoleic acid (omega-6), alpha-linolenic acid (omega-3), and some vitamins (vitamins E, D, and A). Polyphenols and terpenoids, in particular, present in hemp seeds, provide antimicrobial, antioxidant, and anti-inflammatory properties. This review examines the scientific literature regarding hemp seeds’ physicochemical and nutritional characteristics. The focus is on those characteristics that allow for their use in the food industry, aiming to transform ordinary food products into functional foods, offering additional benefits for the body’s health. Innovating opportunities to develop healthy, nutritionally superior food products are explored by integrating hemp seeds into food processes, promoting a balanced and sustainable diet.

## 1. Introduction

In 2020, soy and wheat represented the predominant sources of plant-based protein, contributing significantly, at 57.6% and 36.8%, respectively, to the total food market [1]. However, these two sources, fundamental for human nutritional needs, are increasingly associated with allergenic issues [2]. Both soy and wheat rank among the top eight food allergens, accounting for 90% of all known food allergic reactions [3,4]. The problem intensifies notably with wheat, posing a significant concern for approximately 109 million individuals worldwide, constituting 1.4% of the global population affected by celiac disease—an autoimmune disorder linked to the ingestion of gluten [5]. In light of these challenges, a significant shift in consumer food preferences is highlighted by the growing trend (23%) of limiting meat consumption among the global population [6,7].

This evolution has led to a substantial increase in the consumption of vegan and vegetarian products [8], considered sustainable lifestyle options [5]. Therefore, industrial hemp (*Cannabis sativa* L.) becomes an attractive source of plant protein [1,9], as it does not trigger food allergies and is safe for individuals with celiac disease [10]. Hemp stands out as a renewable resource [3] with promising potential [11], attracting interest due to its versatile use [3], rapid growth [12], low capital investment [13] compared to other crops, and positive environmental impact [5]. Cultivated for thousands of years in China, Asia, and later in Europe, industrial hemp (*Cannabis sativa* L.) was primarily used in the textile industry. However, recent studies have highlighted the significant nutritional (Figure 1), economic, and social importance of the hemp plant (*Cannabis sativa* L.) [12,14,15].

Despite its numerous advantages, certain aspects require further investigation, such as antinutritional factors [16,17] and the accumulation of heavy metals in the soil during the phytoremediation process [12]. The sustained interest in hemp in the food industry has grown, since the approval of strains with a tetrahydrocannabinol (THC) content below 0.3% in 1996 [18], facilitating the cultivation of this plant in European Union countries [3]. As a result of these aspects, it is estimated that by 2025, the global market for industrial hemp (*Cannabis sativa* L.) will reach a value of USD 26.6 billion [19]. Currently, the largest hemp-producing countries in Europe include France, the United Kingdom, Hungary, and Romania [17].

Previous studies, carried out in the last five years, have reviewed the conventional and innovative extraction protocols of bioactive compounds [3,20], the evaluation of the quality of protein from hemp seeds [21], therapeutic effects [22], industrial uses [5,15,17,23,24], the characterization of hemp oil [25], the sustainability of industrial hemp [11,14,26], and the nutritional and chemical composition of hemp seeds [3,27,28].

Considering that research registers continuous progress, the nutritional, technological, and functional aspects of hemp seeds (*Cannabis sativa* L.) must be systematically reviewed for a better understanding of them in developing the potential of this plant to enhance dietary profiles and human health.

Therefore, this paper has been designed as a comprehensive review of scientific sources providing relevant information on the physicochemical and functional characterization of hemp seeds (*Cannabis sativa* L.), including nutritional, functional, and technological aspects. The novelty of this work consists of reviewing those characteristics that enable their use in the food industry, aiming to transform ordinary food products into functional foods, providing additional health benefits to the body. This study may also be applicable because of the current interest in the innovative opportunities to develop healthy, nutritionally superior food products, which are explored by integrating hemp seeds into food processes, promoting a balanced and sustainable diet.

The research was conducted in scientific databases such as PubMed, ScienceDirect, Google Scholar, Web of Sciences, Food Science and Technology Abstracts, MDPI databases, Springer, Wiley, Scopus, and others relevant to food and nutrition. Online resources, books, and research reports were consulted for comprehensive information coverage. Sources that do not provide updated data or align with the objectives of the article have been excluded. Extracted data were categorized into distinct sections, covering the physical aspects of hemp seeds (*Cannabis sativa* L.); high-value compounds including proteins, lipids, carbohydrates, vitamins, and minerals; biologically active compounds (terpenes, flavonoids, phytosterols, carotenoids, and phytocannabinoids) that are found in these seeds that reveal their bioavailability; hemp seed-based functional foods; and value-added food products. Additionally, the antinutritional factors influencing these aspects in the food industry’s technological processes were examined.

## 2. Physical Characteristics of Hemp Seeds (*Cannabis sativa* L.)

Hemp seeds, obtained approximately three to four months after planting [14,23], represent fundamental elements of the hemp plant (*Cannabis sativa* L.). The physical properties of the hemp seeds (Figure 2), as detailed in Table 1, offer measurable characteristics that contribute to defining the species. However, they do not provide information about the plant’s maturation process [13,14,29].

Hemp seeds, covered by a thin pericarp with two layers, have an endosperm and two cotyledons inside. They constitute a valuable energy source, providing 500–600 Kcal/100 g of product, according to data provided by the U.S. Department of Agriculture (USDA) [30]. According to this information, this detailed analysis makes essential contributions to understanding the physical properties and chemical composition of hemp seeds, paving the way for exploring and optimizing the utilization of this resource in the context of food and nutrition [22].

## 3. Nutritional Characterization of Hemp Seeds (*Cannabis sativa* L.)

### 3.1. High-Value Compounds from Hemp Seeds

#### 3.1.1. Hemp Seed Proteins and Their Technological and Functional Characterization

In assessing the nutritional value of proteins derived from hemp seeds (*Cannabis sativa* L.), addressing multiple aspects is imperative, including amino acid composition, digestibility, and bioavailability [22]. Whole hemp seeds, containing approximately 25% of easily digestible proteins, present a notable nutritional advantage, due to the absence of protease inhibitors. By removing the shell [31], the hulling process reduces or eliminates antinutrients, dietary fibers, and color pigments, thus optimizing the use of proteins and increasing the protein content, which is associated with improved digestibility [24].

In the case of hemp seeds, the protein-rich endosperm contrasts with the pericarp, which has a lower protein content and represents an important aspect to investigate in protein extraction and utilization [32]. Removing components (such as oil) with relatively low concentrations of proteins from hulled seeds allows for the production of products with a protein content approximately 1.5 times higher than that of whole seeds [32].

Studies have highlighted that, after oil extraction [13], the amino acid content in the resulting protein powder can be significantly higher compared to whole hemp seeds, a fact attributed to the removal of non-essential anatomical parts [3,32]. The subsequent process of obtaining hemp protein isolate (HPI) demonstrates the extended potential of this protein source [33], opening perspectives for various applications in the food industry and from a nutritional standpoint [19]. According to previous studies, HPI has been identified as having multiple bioactivities, including antioxidant [32] and anti-inflammatory properties [19]. Additionally, HPI has been reported to have hypocholesterolemic effects [28], suggesting its potential contribution to maintaining optimal cholesterol levels in the body.

Hemp seeds stand out in human nutrition due to their remarkable protein content, which is approximately 32 g (Table 2). A crucial aspect of the hemp seed protein profile is the presence of edestin and albumin, two distinct types of proteins that contribute to the exceptional nutritional value of these seeds [34].

Edestin, a storage protein categorized in the legume family, is predominantly found in its 11S globulin form, constituting between 60% and 80% of the total protein in hemp seeds [35]. Particularly valuable, edestin stands out as an excellent source of amino acids [36] with high bioavailability, according to conducted research [33,37].

Simultaneously, albumin, a related globular protein known as 2S, completes the remaining protein fraction of hemp seeds [35]. Approximately 25% of the total reserve proteins come from this protein category. Some recent studies [38,39] have highlighted that the edestin content in different hemp seed protein isolates [33] varies between 70% and 82% of total proteins, while albumin represents approximately 25% of the total seed proteins [35].

These proteins feature a complete amino acid profile. Table 3 presents data from the literature on the amino acid content [36] for whole hemp seeds, hulled seeds, protein powder, and protein isolates derived from hemp seeds (*Cannabis sativa* L.) [3]. This comprehensive analysis provides a detailed perspective on amino acid composition [36] and the potential nutritional [2] and bioactive [14] benefits of hemp proteins, thereby consolidating the role of hemp seeds as a promising source in human nutrition [40].

Recent research [1,22] has emphasized that the protein fraction of hemp seeds represents an easily digestible nutritional source, exhibiting a rich profile of essential amino acids necessary for infants [36]. Detailed analyses of the amino acid profile have confirmed that hemp seed proteins contain all essential amino acids [41] the human body requires [8], with glutamine being the most abundant amino acid, followed by arginine. In addition to focusing on essential amino acids [42], evaluating the benefits of non-essential amino acids, such as arginine, is essential. Previous studies have highlighted that arginine [43], present in hemp seeds, plays a significant role in the production of nitric oxide (NO). This compound has vasodilatory properties, facilitating the dilation of blood vessels and thus contributing to improved blood circulation [14].

Moreover, a significant amount of methionine can enhance the nutritional quality of plant-based foods. Thus, the hemp seed profile is comparable to casein, except for lysine, which becomes the first limiting amino acid in hemp seed proteins. This finding is significant in the context of the nutritional needs of infants up to the age of 5, a period during which adequate lysine intake is essential [14].

This perspective on the protein composition [32] of hemp seeds provides a solid foundation for understanding the essential role of edestin and albumin in the contribution of hemp seeds to human nutrition and suggests their potential in developing high-quality foods and dietary supplements [14,44]. Differences in researchers’ results may be associated with variations in the hemp strains used in the analysis, extraction methods, and conducted analyses [40,45].

Comparing the nutritional profile of hemp seeds with those of other natural sources highlights their diversity and significant potential (high content in proteins, lipids, and some vitamins and minerals) [46,47]. Hemp milk, derived from whole seeds, has become an appealing alternative to soy and nut milk, not only offering a nutritional source, but also potential health benefits, including cholesterol reduction [48,49] (Table 2).

**Table 2 molecules-29-02097-t002:** Comparison of nutritional components between hemp seeds and other natural sources.

Product	Calories (kcal)	Proteins (g)	Fats (g)	Carbohydrates (g)	mg/100 g													
					Vit. E	Vit. D	Vit. B1	Vit. B2	Vit. B3	Vit. B6	Ca	Fe	P	Mg	Zn	K	Cu	References
Hemp seeds	567	30	50	10	1	0.1	1	1.1	2.5	0.3	70	8	1667	700	10	1200	1.6	[50,51,52,53]
Hulled hemp seeds	553	32	49	9	1	0.1	1.3	0.3	9	0.6	70	8	1650	700	10	1200	1.6	[51,52,53]
Sunflower seeds	571	21	50	18	0.7	0	0.32	0	6.9	1.34	116	4.37	732	302	5.58	657	1.88	
Pumpkin seeds	533	23	43	17	0.2	0	0.03	0	4.2	0.14	37	8.36	1150	500	6.34	691	1.22	
Peanuts	567	26	49	16	4.3	0	0.6	0.1	12	0.4	92	5	376	168	3.3	705	1.1	
Soybean	446	36	20	30	0.9	-	0.9	0.9	1.6	0.4	277	16	704	280	5	1797	2	
Wheat	339	14	2.5	71	1	0	0.4	0.1	6.7	0.4	34	3.5	508	144	4	431	0.6	
Oat	379	13	6.5	68	0	0	0.4	0.2	0.1	0.1	52	4.2	410	138	3.6	362	0.4	
Quinoa	368	14	6.1	64	2.5	0	0.4	0.3	1.5	0.5	47	4.5	457	197	3.1	563	0.6	
Corn	365	9.4	4.7	74	0.5	0	0.4	0.2	3.6	0.6	7	2.7	210	127	2.2	287	0.3	
Lentils	352	25	1.1	63	0.5	0	0.9	0.2	2.6	0.5	35	6.5	281	47	3.3	677	0.8	
Peas	81	5.4	0.4	14	0.1	0	0.3	0.1	2	0.2	25	1.5	108	33	1.2	244	0.2	
Beans	333	23	0.9	60	0.2	0	0.4	0.2	0.5	0.4	240	10	301	190	4	1795	1	
Hemp milk	130	4	3	20	0	0.02	-	-	-	-	520	2.7	538	80	0.1	-	-	[40,51,53]
Soy milk	80	7	5	4	0	0.05	0.3	0.2	0.4	0.03	455	7.2	188	63	0	-	-	
Cow milk	150	8	8	11	0	0.04	0.3	1.2	0.3	0.3	377	0.18	275	25	0	-	-	

In the food industry, proteins are widely used in various forms, due to their nutritional value and technological and functional properties that can influence the behavior and performance of ingredients [19,21,34,50]. A recent study from 2022 [3] emphasizes the importance of protein characteristics and their functionalities [33], which are profoundly influenced by various factors such as genotype, chemical structure, and complex interactions with other components of the food matrix, such as water, carbohydrates, and lipids [43].

Production processes, including dehulling and isolation (micellization), are key elements that can affect the performance of hemp proteins in food products [22] For instance, some studies highlight that dehulling techniques [31] have a limited impact on hemp protein subunits [47,54]. In contrast, pH can play a significant role in the accumulation of edestin through isoelectric precipitation during the alkaline extraction process (AE-IEP) [35,55].

According to some studies [21,39,56], the secondary structure of hemp proteins is pH-dependent and can influence their solubility. At pH 3.0, α-helix dominates, followed by β-sheets, whereas at pH 7.0, a lower amount of α-helix is observed. At neutral pH, hemp protein is dominated by β-sheets (41–47%) and α-helix (19–28%), showing similarities with other protein-rich legumes such as beans and peas [37].

Solubility, thermal stability, and emulsification properties are important characteristics of proteins in the food industry that influence technological processes and the quality of end products [48]. Understanding these aspects contributes significantly to optimizing technological processes and improving the quality of food products containing hemp proteins [19,37,42].

The water solubility of HPI ranges from 13 to 17% at a neutral pH and reaches 74–80% at an alkaline pH of 9 [33]. This alkaline solubility contributes significantly to the functionality of proteins in various technological applications [19,21].

Thermal stability is necessary to maintain nutritional quality [53]. The maximum denaturation temperature of hemp proteins ranges from 89 to 96 degrees Celsius [48].

Emulsification is influenced by the rate and quantity of protein adsorption at the oil–water interface. Hemp protein is known for its amphiphilic properties, having both hydrophilic and lipophilic characteristics. However, its emulsifying property is considered relatively weak, mainly due to its low solubility, comparable to rapeseed protein [41].

Protein foaming is essential in producing various foods and this process depends on protein adsorption at the air–water interface, their configuration, and the formation of cohesive viscoelastic films. Protein concentration, solubility, and hydrophobicity can significantly influence this process [21,23,49]. Water Holding Capacity (WHC) and Oil Holding Capacity (OHC) [13] represent valuable aspects of protein functionality in food systems. These properties contribute to maintaining the moisture, freshness, and optimal texture of foods [48].

Gelation represents the minimum protein concentration required to form a gel. This process depends on structure, concentration, pH, and temperature. For HPI [35], this concentration is 22%, while for hemp protein isolate obtained using micellization (HMI), it is 12% [33], indicating significant variations between the two different types of hemp proteins [33,37].

The digestibility of proteins from hemp seeds and their effects on human health are the subject of intense research, being approached from various perspectives. Hemp seeds, rich in protein (21–32%), have been compared to legumes such as lentils or beans, surpassing wheat nutritionally. The content of tryptophan and lysine, identified as limiting amino acids, reduces the digestibility of hemp seed proteins compared to soy. However, hemp seed protein has a good digestibility rate, ranging from 84 to 86% for whole seeds and up to 83–92% for hemp seed flour. Digestibility is measured using the Protein Digestibility Corrected Amino Acid Score (PDCAAS) method.

However, the PDCAAS score (approximately 1.0) indicates that there are still challenges in obtaining products with optimal digestibility. The increasing trend in the consumption of hemp protein-containing foods highlights their importance in the food industry [2,14,36]. Different antinutritional compounds and processing methods can influence the digestibility of hemp seeds. Removing the shells before protein extraction and heat pretreatment of the protein isolate have been some of the attempted approaches to improve digestibility [56].

The allergenicity of hemp seeds is low [3], which makes hemp protein isolate (HPI), containing approximately 86% edestin [35], a promising option for hypoallergenic foods. The protein lacks known allergens derived from hemp, thus supporting its safe utilization for individuals with allergies [21].

The bioactive activities of hemp proteins have also been studied. Results have indicated that the digestion of hemp proteins can generate bioactive peptides with antioxidant [2], antihypertensive [14], antimicrobial [29], and cytomodulatory benefits. Peptides resulting from the hydrolysis of hemp proteins have demonstrated hypertensive [49], hypocholesterolemic [48], and hypoglycemic activity, indicating their potential in managing cardiovascular risks [57] and other conditions [49,57].

Moreover, hydrolyzed hemp proteins [14] have shown neuroprotective effects [22], inhibiting acetylcholine activity, which may have implications in preventing neurodegenerative diseases like Alzheimer’s disease (A.D.). Recent studies have also highlighted the ability of hemp proteins to bind zinc, improving its bioavailability [58,59].

#### 3.1.2. Hemp Seed Lipids—An Excellent Source of Essential Fatty Acids

The oil extracted from hemp seeds (*Cannabis sativa* L.) is the subject of in-depth research, due to its rich composition of beneficial substances and potential effects on human health. Hemp seeds contain around 32.21% oil [13], but this percentage can vary depending on hemp varieties and cultivation and processing conditions. Sensory-wise, hemp oil has a light-yellow color, a mild taste, and a delicate nutty flavor. The green color of freshly obtained oil is due to the natural presence of chlorophyll in mature seeds [13,23].

The lipid profile of hemp oil is remarkable, containing polyunsaturated fatty acids, such as linoleic acid (omega-6) and alpha-linolenic acid (omega-3) [22]. Additionally, hemp oil has a high content of essential fatty acids such as stearidonic acid and gamma-linolenic acid (GLA) [14,60]. A significant aspect of hemp oil is its tocopherol content, with γ-tocopherol [13] being the predominant component. The tocopherol content makes it a valuable source of antioxidants, protecting lipids against oxidative degradation. The oil extraction process involves cold-pressing the seeds at controlled temperatures, maintained between 45 and 50 °C, to preserve the nutritional quality of the essential fatty acids. This technique is similar to that used for other oilseeds and complies with Codex Alimentarius standards [61]. Further refining can eliminate undesirable substances and improve the color, but there is a risk of losing essential vitamins and tocopherols [62].

An important aspect of proper hemp oil storage is maintaining the temperature between 8 and 15 °C, to ensure the stability of its nutritional quality [50]. Thus, research on hemp oil emphasizes its complex composition and the importance of processing and storage technologies, to maximize its nutritional and functional qualities [13].

Polyunsaturated omega-3 fatty acids—alpha-linolenic acid (ALA) and omega-6-linoleic acid (LA) [24]—are essential for the human body, as they cannot be synthesized by it and must be obtained through diet. However, it is crucial to maintain an adequate ratio between these fatty acids, since omega-6 can have pro-inflammatory effects, while omega-3 has anti-inflammatory effects [13].

A recent study on hemp oils indicated that hemp seeds represent an excellent source of essential fatty acids [29], containing, on average, over 70% polyunsaturated fatty acids (PUFAs) [14], with an optimal ratio of 3:1 between linoleic acid (omega-6) and alpha-linolenic acid (omega-3), according to the European Food Safety Agency recommendations [63].

The chemical composition of hemp oil and parameters like acidity, peroxide value, and absorbance are essential for its qualification for human consumption. Differences in hemp seed harvesting and processing methods can significantly influence the content of bioactive substances [14]. However, the significant amount of PUFAs may increase the susceptibility of hemp seeds to oxidation under heat, light, or air, thereby affecting food quality and shelf life [13,21].

The hemp oil extraction process cannot completely exclude the unwanted flavors resulting from its oxidation [22]. Prevention strategies, such as storage at low temperatures, should be applied to hemp seeds from harvesting to ensure the quality and subsequent functionality of food ingredients like proteins and oils [3].

The high antioxidant capacity [19] of hemp oil, comparable to extra virgin olive oil [64], adds a beneficial aspect to its use in various applications. Phytosterols and iodine value contribute to the health benefits of hemp oil, protecting against cardiovascular diseases [54] and supporting the health of blood vessels and the nervous system. In foods, cosmetics, medications, and industrial applications [23], hemp oil is a valuable ingredient in human health and its versatility [2,13].

#### 3.1.3. Carbohydrate Content of Hemp Seeds and Their Nutraceutical Effects

Studies [50,65] on the nutritional composition of hemp seeds have highlighted a significant variation in the total carbohydrate content, ranging between 20% and 30%. Analyzing the nutritional value of hemp seeds, it is observed that the values for the total carbohydrate content in whole seeds are comparable to those found in whole flax seeds. Statistical data indicate approximately 34.4 ± 1.5 g/100 g for hemp seeds and 29.2 ± 2.5 g/100 g for flax seeds [48].

Dietary fibers are an essential component of human nutrition, defined as the edible portion of plant material [5] that resists enzymatic digestion in the small intestine and undergoes partial or complete fermentation in the large intestine. Dietary fibers encompass various components such as cellulose, hemicellulose, pectin, gums, mucilages, and lignin. Hemp seeds are distinguished by a significant content of dietary fibers, representing a rich source of carbohydrates, with a predominant proportion of insoluble dietary fibers (IDFs) [18,48].

The health benefits of dietary fibers are well-known and associated with positive effects on the digestive tract, cholesterol reduction, and blood glucose level management [48,49]. The fermentation of fibers in the colon generates short-chain fatty acids (SCFAs) [14], which play beneficial roles both at the intestinal and systemic levels. Hemp seeds can, thus, be integrated as functional ingredients in foods, contributing to meeting the recommendations for dietary fiber intake, especially considering their reduced consumption in Western countries.

However, it is important to carefully evaluate industrial processing methods, as they can influence the fiber composition, and seed hulling can alter the nutrient proportions. Assessing the balance between optimal fiber intake and interference with the absorption of other nutrients is crucial, emphasizing maintaining a balanced and healthy diet.

#### 3.1.4. Hemp Seeds’ Vitamins and Minerals with High Nutritional Potential

Hemp seeds (*Cannabis sativa* L.) stand out due to their rich vitamin content, which has the potential to meet or even exceed the recommended daily allowance (RDA) for essential vitamins (Table 3). According to some research [14,50], just 50 mg of hemp seeds or 15 g of hemp seed oil [13] provide over 100% of the RDA for vitamins E, D, and A, thereby meeting the nutritional requirements of an adult [66].

**Table 3 molecules-29-02097-t003:** Amino acid content for whole hemp seeds, hulled hemp seeds, hemp protein powder, and hemp seed isolates.

	A.A.	Ala	Arg	Asp	Cys	Glu	Gly	His *	Ile *	Leu *	Lys *	Met *	Phe *	Pro	Ser	Thr *	Trp *	Tyr	Val *	Ref.
Product																				
**Whole seeds**		1.28	3.10	2.78	0.41	4.57	1.14	0.71	0.98	1.72	1.03	0.58	1.17	1.15	1.27	0.88	0.20	0.86	1.28	[43]
		0.90	2.48	2.16	0.33	3.59	0.93	0.56	0.83	1.31	0.75	0.49	0.89	0.82	0.99	0.71	0.43	0.57	0.97	[22]
		1.53	4.55	3.66	0.67	6.27	1.61	0.96	1.61	2.16	1.28	0.93	1.45	1.6	1.71	1.27	0.36	1.26	1.78	[30]
		0.96	2.28	2.39	0.41	3.74	1.06	0.55	0.80	1.49	0.86	0.56	1.03	0.90	1.19	1.01	0.23	0.68	1.14	[46]
**Hulled seeds**		1.52	4.55	3.66	0.65	3.74	1.61	0.97	1.29	2.14	1.26	0.94	1.43	1.62	1.70	1.27	0.38	1.28	1.78	[41]
		0.45	1.36	1.09	0.20	1.88	0.48	0.29	0.38	0.64	0.38	0.28	0.43	0.47	0.51	0.38	0.11	0.37	0.53	[49]
		4.43	13.2	10.8	1.9	18.7	4.70	2.83	3.76	6.24	3.68	2.74	4.17	4.73	4.96	3.70	1.11	1.11	5.19	[21]
**Protein powder**		1.61	3.90	3.65	0.69	6.03	1.66	0.93	1.44	2.34	1.31	0.88	1.62	1.59	1.73	1.34	0.39	1.15	1.90	[46]
		4.25	11.9	11.17	1.62	17.5	4.86	3.48	3.81	6.88	4.21	1.94	4.74	4.98	5.55	3.77	1.05	2.79	5.34	[21]
**Hemp seed protein isolate**		5.2	12.0	10.9	1.7	17.2	4.9	2.9	4.1	6.9	4.2	2.4	4.7	4.7	5.3	3.3	0.8	3.5	5.3	[38]
		3.59	8.67	8.91	1.84	17.6	3.78	2.52	4.07	6.56	4.54	1.77	4.43	4.55	5.09	3.87	0.92	3.74	4.07	[31]

* Essential amino acids. Ala: Alanine; Arg: Arginine; Asp: Asparagine; Cys: Cysteine; Glu: Glutamate/Glutamine; His: Histidine; Ile: Isoleucine; Leu: Leucine; Lys: Lysine; Met: Methionine; Phe: Phenylalanine; Pro: Proline; Ser: Serine; Thr: Threonine; Trp: Tryptophan; Tyr: Tyrosine; Val: Valine.

Vitamin E, renowned for its crucial antioxidant role [19] contributes to cell protection against oxidative stress. Vitamin D, essential for bone health and the immune system, and vitamin A, beneficial for visual function and skin integrity, are also significant in hemp seeds. The B vitamin group, essential for proper nervous system function, is well-represented in hemp seeds. For instance, vitamin B1, 1.3 mg/100 g, and vitamin B9, 18 mg/100 g, support nervous system health and cell division processes [67,68].

This vitamin richness gives hemp seeds significant nutritional potential that can enhance daily nutrition. Incorporating these seeds into the diet can help to ensure an adequate intake of essential vitamins, providing multiple health benefits for overall health and biological system functioning [2].

Minerals represent essential elements for numerous physiological processes, playing a crucial role in maintaining the body’s acid–base balance [50,54]. However, information regarding mineral elements and their bioavailability [37] in hemp seeds remains limited. Phosphorus is abundant in hulled seeds (1.1 g/100 g) [54], with a higher content than in whole seeds. Other minerals like potassium, magnesium, and zinc are also present in significant quantities (Table 3). Whole seeds contain more calcium, manganese, and copper than hulled seeds. Iron, vital for the body, is found in similar concentrations in both whole and hulled seeds (8 mg/100 g), while sodium remains below 5 mg/100 g [54]. One important aspect to consider in interpreting the mineral content of seeds is the presence of phytates, especially in hulled seeds (4 g/100 g). These compounds can compromise the absorption of iron and zinc, with phytate/Fe and phytate/Zn molar ratios exceeding 20 and 15, respectively. Nevertheless, hemp seeds significantly contribute to the RDA of manganese (>5 mg/100 g) and copper (>1.4 mg/100 g), providing approximately 100% and 50% of the RDA in a 30 g serving [24,54].

It is important to emphasize that minerals are more resistant to processing than vitamins but can undergo changes if exposed to light, air, or heat. Therefore, hemp oil should be avoided when cooking. It is recommended to consume vitamins and minerals in their natural form to ensure optimal health benefits, with vitamin E preventing diabetes; B vitamins supporting the nervous system; and minerals like zinc, selenium, or copper acting as antioxidants [2,50,67].

### 3.2. Biologically Active Compounds

Secondary metabolites constitute an essential component of the defense response of the hemp plant (*Cannabis sativa* L.) to biotic or abiotic stress. These bioactive compounds, secreted by seeds, comprise a wide range of compounds, including terpenes, phenolic compounds, alkaloids, phytocannabinoids, tocopherol isomers [13], beta-tocopherol, gamma-tocopherol, alpha-tocopherol, and delta-tocopherol, with a preference for the gamma-tocopherol derivative in significant quantities [54]. It is crucial to highlight that variations in the composition of these secondary metabolites can be influenced by cultivation conditions, providing a distinct fingerprint of different production regions.

Through the implementation of multivariate analysis, it is possible to distinguish hemp seeds originating from various regions efficiently, thus decoding the geographical characteristics of raw materials. This diversity in the composition of secondary metabolites reflects plant adaptations to the environment and opens doors to a deeper understanding of the complex interactions between plants and environmental factors [69].

Consequently, a detailed analysis of these secondary metabolites provides a more comprehensive perspective on the geographical influences on hemp plants and their unique biochemical resources [17].

#### 3.2.1. Terpenes

Terpenes, natural chemical substances found in plants, including cannabis and hemp, play an essential role in providing the scent and flavor of plant-based foods. Oil extracted from hemp seeds is distinguished by its high concentration of terpenes, representing one of the richest sources of these phytochemical substances [70]. Over 85 volatile compounds have been detected in hemp seed oil, belonging to the chemical classes of terpenes, heterocyclic compounds, hydrocarbons, alcohols, furans, ketones, pyrazines, pyrroles, aldehydes, acids, and esters [21,71,72]. The terpenes found in the highest concentration in hemp seed oil, which are considered key contributors to the unique aroma with odor activity values, are presented in Table 4.

Studies have highlighted that terpenes in hemp seed oil play significant roles in protecting it against oxidative stress [19], due to their ability to scavenge free radicals [73]. Furthermore, research suggests terpenes can enhance well-being within a healthy lifestyle. The expected benefits include reducing physical pain and tension, balancing mood and appetite control, improving cognitive functions, and supporting sleep and anxiety management [2,74].

Among the terpenes in hemp seed oil, β-myrcene stands out for its significant anti-inflammatory and anticatabolic effects in human chondrocytes [22]. This suggests its ability to slow down or even halt cartilage destruction and osteoarthritis progression. However, it is important to note that there is limited clinical evidence regarding the effects of specific components from *Cannabis sativa* L. on managing osteoarthritis-associated pain, which requires further investigation. Also, β-caryophyllene has been widely investigated and highly regarded for its low toxicity and considerable safety profile. The existing research indicates that β-caryophyllene can enhance insulin secretion, sensitivity, and glucose uptake, while decreasing glucose absorption. Furthermore, it lowers triglyceride and cholesterol levels, promotes fatty acid oxidation, and maintains lipid homeostasis, thus manifesting hypolipidemic effects [75,76].

#### 3.2.2. Flavonoids

Flavonoids, a subclass of phenolic compounds such as flavanones, flavonols, flavanols, and isoflavones, have been identified as being among the most abundant compounds present in hemp seeds (*Cannabis sativa* L.) [19,77]. These compounds have been recognized for their beneficial properties, including anticancer, anti-neuroinflammatory [20], antioxidant [19], and general anti-inflammatory activities [22]. Studies have indicated that some flavonoids also exhibit arginase inhibition activity [77], thereby contributing to increasing nitric oxide (NO) levels, a signaling molecule in the cardiovascular system that plays a significant role in maintaining blood pressure homeostasis [57].

Another point of interest in the food industry’s technological processes is the color of products derived from hemp seeds. Color represents a significant challenge, because visual appearance is essential for consumer acceptance [19,21] HPI obtained from whole seeds is dark green, generated by the presence of phenolic compounds and chlorophyll [33]. These water-soluble pigments participate in the formation of dark colorants [5] through polymerization and oxidation reactions during the alkaline extraction–isoelectric point precipitation (AE–IEP) process [77,78].

However, dehulling, protein processing, and extraction methods can influence the color of the final products. Removing hulls during dehulling can lead to losing the dark green color, offering a more appealing shade for consumers. These technological adjustments can contribute to improving the color of hemp seed protein isolate, making the product more attractive to consumers [21,31,33].

#### 3.2.3. Phytosterols

Phytosterols, compounds with a structure like cholesterol [49], are essential in hemp seeds (*Cannabis sativa* L.). Unlike cholesterol, phytosterols cannot be synthesized by the human body and can be found exclusively in plants. The primary beneficial characteristic of phytosterols is their ability to modify cholesterol solubility in the intestine, thereby reducing its absorption [49]. This process occurs by excluding cholesterol from the lipid fraction, mediated by competition in lipid micelles. Although the specialized literature is limited, it has been found that the concentration of phytosterols in hemp seeds is significant, approximately 280 mg/100 g in oil and approximately 124 mg/100 g in the whole seed. The dominant isoform of phytosterol in hemp is beta-sitosterol, with concentrations ranging from 190 to 54 mg/100 g, with the highest values being recorded in seed oil [48].

One notable aspect of beta-sitosterol is its ability to reduce cholesterol levels [48], which is beneficial to maintaining cardiovascular health [54]. Additionally, the anti-inflammatory effects of phytosterols have been described [22]. Hemp and pistachios seem to be one of the richest natural sources of beta-sitosterol. The unsaponifiable fraction of hemp seeds, representing less than 2% of oil, includes, in addition to phytosterols, sterols, tocopherols, and water-soluble vitamins, thereby contributing to the overall benefits of hemp consumption for human health [2,13].

By comparison, olive oil contains half the unsaponifiable fraction found in hemp seed oil, and beta-sitosterol accounts for approximately 15% of this fraction [2]. These findings suggest that phytosterols from hemp seeds can play a significant role in maintaining cardiovascular health and promoting a healthy diet [57].

#### 3.2.4. Carotenoids

Carotenoids are highlighted in hemp seeds (*Cannabis sativa* L.), with lutein, which is the most abundant, ranging from 1.4 to 3.4 mg/100 g in the whole seed [2]. Lutein and zeaxanthin accumulate in macular cells, protecting against light-induced oxidative stress. This indicates the potential of hemp seeds to contribute to maintaining ocular health and reducing the risk associated with macular degeneration.

Hemp oil complements this remarkable nutritional composition, offering approximately 80 mg/100 g total tocopherols [13], lipid-soluble compounds with strong antioxidant properties [19]. These plant-derived tocopherols represent an essential shield against oxidative stress and contribute to maintaining cellular health. Furthermore, hemp seeds contain molecules such as sativamides, which have been associated with potential benefits in neurodegenerative diseases, including Parkinson’s disease and Alzheimer’s disease (AD) [34]. These findings open fascinating perspectives in researching hemp compounds for developing treatments and preventing neurological conditions.

Therefore, the bioactive compounds in hemp seeds, with antioxidant [19] and anti-inflammatory effects [22], propose hemp oil as a promising alternative to current diets. Also, due to its composition, it is suitable for vegetarian, vegan, or gluten-free diets, highlighting this food’s versatility and nutritional value [34].

#### 3.2.5. Phytocannabinoids Identified in Hemp Seeds

According to recent studies, over 120 phytocannabinoids have been identified in the hemp plant, among which Δ9-tetrahydrocannabinol (Δ9-THC) and cannabidiol (CBD) are considered the most important regarding therapeutic impact [14,56]. However, hemp seeds are high in lipids, with significant health benefits due to the presence of essential fatty acids, but contain no more than trace amounts of phytocannabinoids such as cannabidiol (CBD) and Δ9-tetrahydrocannabinol (Δ9-THC), the psychoactive compound found in *Cannabis sativa* [79]. The Δ9-THC content in hemp seeds is well below the legal threshold of 0.3% in the USA, and 0.2% in the EU, making them non-psychoactive and safe for consumption [80]. During the harvesting process, hemp seeds can be contaminated on the outer surface through the sticky resin, rich in cannabinoids secreted by the numerous glandular trichomes present on the bract [81,82]. However, through the appropriate preconditioning of seeds (washing/cleaning them), the obtained hemp oil will contain only traces of phytocannabinoids [83]. In a study carried out by Citti et al. (2019) on the cannabinoid profile of ten commercially organic hemp seed oils, using a liquid chromatography method coupled with high-resolution mass spectrometry, 32 phytocannabinoids were identified [83].

Yang et al. (2017) investigated three brands of hemp seeds, using four different procedures of phytocannabinoid extraction and quantified total CBD (microwave extraction, Soxhlet extraction, sonication, and supercritical fluid extraction). According to their study, the amount of phytocannabinoids extracted depended on the extraction procedure. Therefore, the total concentration of CBD varied from 217 to 227 μg/g, the highest concentration being recorded in the extracts obtained using the Soxhlet extraction [79]. Another study that investigated the evaluation of CBD and Δ9-THC content in hemp seed oil recorded a total CBD content of less than 5 mg/kg. The investigated hemp seed oils did not have a high total content of neutral cannabinoid Δ9-THC < 2 mg/kg [Citti, 2018] [84].

CBD, with vast therapeutic potential, is being researched for possible benefits in treating painful conditions such as fibromyalgia, migraines, and irritable bowel syndrome, presumed to be linked to an endocannabinoid deficiency [56]. Despite strict regulations, research continues to reveal the therapeutic potential of CBD, bringing promising perspectives for developing innovative treatment options [3,56,85].

Table 5 highlights the discussed health functions of hemp seed’s nutrients.

### 3.3. Antinutritional Factors

Antinutritional factors in hemp seeds (*Cannabis sativa* L.) represent an important component present in plant-based proteins [1], which can significantly influence protein digestibility [41] and nutrient bioavailability [21].

From the composition of these seeds, trypsin inhibitors, phytic acid, cyanogenic glycosides, condensed tannins, and saponins, predominantly located in the cotyledon fractions, have been implicated in limiting the optimal utilization of plant proteins in food systems [17]. For example, trypsin inhibitors can reduce protein digestibility [41], while phytic acid can affect the bioavailability of essential nutrients [37]. The average phytate content in whole hemp seeds has been evaluated at 2.80 g/100 g, signaling a possible interference with adequate nutrient absorption [8].

However, recent research has revealed surprising perspectives on these antinutritional factors. The absence of trypsin inhibitory activity has suggested that seed protein can be used to enhance the nutritional quality of plant-based foods [44].

Fermentation and germination processes [19] have demonstrated efficacy in inhibiting these factors, paving the way for improving organoleptic characteristics and maximizing the use of proteins from hemp seeds in nutrition. Studies indicate that these strategies can transform antinutritional compounds into beneficial compounds for health. Consequently, although hemp seeds contain antinutritional factors, recent research highlights that fermentation and germination processes can be efficient solutions for the optimal utilization of proteins and nutrients from these seeds [56]. These findings lead to a more efficient exploitation of the nutritional potential of hemp seeds in food systems, thus promoting a healthy and varied diet [11].

## 4. Utilization of Hemp Seeds in Functional Foods

Hemp seeds from *Cannabis sativa* L. (43.60%) are the most utilized hemp plant component in the food industry due to their lipid profile and rich protein content, making them an excellent source of nutrition [2]. Also, hemp seeds and their derived products have gained significant popularity among consumers, due to their superior nutritional profile. Hemp seeds can be consumed as whole seeds, shelled, or integrated (hempseed-derived products) into a wide range of food products [11].

Hemp seed oil from *Cannabis sativa* L. has gained commercial popularity, being considered a natural product, with an increasing reputation for dietary use and its medicinal and nutraceutical potential as a dietary supplement [81,86]. Due to the antioxidant and anti-inflammatory properties associated with hemp seed components, several studies have explored their potential in counteracting chronic degenerative diseases characterized by chronic inflammatory and oxidative stress, such as cardiovascular and neurodegenerative diseases [22], atopic dermatitis [87], and red blood cell composition [88]. Overall, the analyzed studies highlight the potential of hemp seeds and their derivatives as beneficial dietary supplements for health [89].

Energy bars are a commonly consumed dietary supplement, especially by athletes and individuals engaged in intense physical activities, to meet the body’s caloric needs [90] or to address insufficient protein intake in underdeveloped regions or nutritionally deficient countries [91]. Results suggest that hemp seeds can be used individually as powder and additives or in a mixture, such as rice/hemp extrudates. Both variants efficiently enrich foods with proteins and improve their characteristics [90,92].

Recent studies support that hemp seed flour is a superior nutritional option compared to wheat flour, with the potential to enhance the nutritional value of food products [3]. In bread production experiments, the use of hemp flour significantly influenced the final product’s quality [93], resulting in decreased volume and increased protein and lipid content, especially at a concentration of 20% [86]. Sensory evaluation indicated decreased sensory perception quality, impacting texture and aroma, especially with 30% and 50% hemp flour additions [93]. Substituting wheat flour with hemp flour in regular bread brought significant positive changes in crumb texture and polyphenol composition, based on the added percentage of hemp flour. Sensory acceptance increased, especially regarding color and aroma [94]. A yeast and mold quantity reduction was observed in hemp flour bread samples, positively impacting the product’s energy value [22]. Shim (2008) patented a process for making bread and pastries from hemp seed oil and hemp seeds, highlighting the potential of this raw material in creating innovative products [95]. Hemp flour has also been used to reduce bread’s gluten content or create gluten-free baked goods [93]. Adding hemp seeds, flour, and proteins to bread production significantly improved the nutritional value of products, with significant increases in fiber and protein content [96]. The results show that hemp flour and proteins can benefit baked product formulations significantly, contributing to improved nutritional quality, sensory acceptability, and adaptability to various types of dough [97].

Due to the natural absence of gluten, hemp seeds are a promising candidate for integration as an ingredient in gluten-free product formulations, aiming to enhance the reduced nutritional value of these products. This is primarily due to a significant increase in total protein, fats, minerals, and total dietary fiber, including soluble and insoluble fractions, with a particular emphasis on the insoluble fraction. These findings highlight the potential of hemp seeds to improve the nutritional profile of gluten-free products [22].

A study conducted by Teterycz et al. (2021) investigated fortifying pasta, by adding hemp flour in varying concentrations ranging from 5% to 40%. This addition improved protein, total dietary fiber, ash, and fat content in pasta composition. Sensory aspects of fortified pasta were well-received by consumers, and THC and CBD levels were considered safe [18].

The impact of replacing traditional flour with hemp seed flour in cookie dough was studied by Farinon et al. (2020). The results showed a significant increase in the general hydration properties of the mixture. Regarding cookies, replacing flour with hemp flour led to a decrease in volume and an increase in hardness, reaching the highest value with cookies enriched with 60% hemp flour. In both cases, cookies and biscuits, the color of the crumb and the cake became darker with increasing levels of hemp seed substitution, positively affecting the sensory acceptability of the product [22]. Wolf et al. (2017) developed a procedure for making brownies with hemp seed oil from *Cannabis sativa* L., establishing an efficient protocol for the analysis and quality control of edible products with cannabinoids [98]. Also, Radočaj et al. (2014) aimed to develop gluten-free foods, focusing on creating biscuits that integrate nutritious hemp flour and decaffeinated green tea leaves [99]. Superior nutritional characteristics, including proteins, crude fiber, minerals, and essential fatty acid properties, provided the product with significant added value and highlighted the potential of hemp flour as a functional ingredient in creating healthy, accessible, and potentially functional food products, especially for individuals with celiac disease [22].

Potato chips are a globally favored snack, produced in two main types, as follows: traditional—obtained by cutting and frying potatoes—and reconstituted—made from a dough with potatoes, starch, emulsifier, and water that is extruded or pressed and then fried. Adding hemp seed flakes influenced the final product’s thickness, density, surface roughness, and composition, by reducing oil absorption during frying. The higher the quantity of hemp seeds, the lower the lipid content. This interaction formed a more compact protein–carbohydrate matrix, reducing oil infiltration during frying [100].

Hemp seed milk from the *Cannabis sativa* L. plant is a popular alternative to cow’s milk, suitable for individuals with lactose intolerance, allergies, or those adopting a vegetarian or vegan lifestyle. Making hemp seed milk at home involves combining water with hemp seeds, resulting in a substitute with a protein level of 0.83 g/100 mL and a significant content of alpha-linoleic acid, an important omega-3 fatty acid. This hemp milk, with fewer calories, proteins, and carbohydrates compared to cow’s milk, but with a similar fat content, offers a creamy texture and an earthy taste. It is versatile and can be used in various preparations such as smoothies, coffee, and cereals, providing a healthy and delicious alternative [92]. By applying high-pressure homogenization (HPH) and pH-change treatment, thermally untreated hemp “milk” can be obtained with stability, physical resistance to oxidative deterioration, and reduced microbial population, indicating its potential as a milk substitute [59]. Furthermore, Ferdouse et al. (2024) have developed a process to obtain hemp “milk” that does not change color and does not develop bitterness during pasteurization [101].

Dabija et al. (2019) introduced hemp seed protein additives into yogurt, aiming to fortify it and create a new daily product with increased nutritional value. Adding hemp seed proteins also influenced the physicochemical characteristics of fortified yogurt, resulting in a reduction in pH value and an increase in acidity. These changes can be correlated with improving the growth process of bacteria present in yogurt, leading to significant positive effects [102,103].

According to Burton et al.’s study (2022a), fermented drinks are obtained from hemp seeds, either containing 3% hemp seeds or through 1:1 (*v*/*v*) mixtures of fermented hemp–soy or hemp–rice beverages, fermented with the addition of probiotics [3].

Proteins from hemp seeds are considered alternative protein sources to animal-origin proteins, potentially contributing to producing acceptable meat analogs [3]. Zając and Świątek (2018) investigated the effects of using whole hemp seeds in liver pâtés, noting a significant improvement in the fatty acid profile, reducing monounsaturated and saturated fatty acids, while increasing polyunsaturated fatty acids (PUFAs), without affecting sensory parameters [22]. Another study [104] evaluated the acceptability and nutritional characteristics of pork products enriched with whole hemp seeds, hulled seeds, hemp flour, and hemp proteins, resulting in significant improvements in protein, ash, and fiber content, reducing oxidation processes in the product. In this case, consumers showed a higher preference for products enriched with hulled hemp seeds.

The food industry is focusing on harnessing the nutritional potential of *Cannabis sativa* L. hemp seeds to develop innovative products. Companies like Allive^®^, Nutiva^®^, Manitoba Harvest^®^, and Canah^®^ have created diversified lines of food products, highlighting the health and nutritional benefits of hemp seeds. Concurrently, brands such as Turn^®^, Cannabia^®^, Mandrin^®^, Coors Light^®^, and Appenzeller Hanfblüte^TM^ are exploring a variety of hemp-based products, including hemp cocktails (Hempfy gin tonic), alcohols (where hemp seeds are used as flavor), lemonades like Hempfy Martini, and HempTea teas, protein blends, infused beers and wines, and coffee-related products, with a niche market focused on the organic products available in specialty food stores. The products are free of genetically modified ingredients, gluten, or added sugar, relying exclusively on plant-based nutritional claims [92]. Also, the CANNUSE database, available at http://cannusedb.csic.es, represents a comprehensive and organized source of information regarding the uses of *Cannabis sativa* L. hemp seeds, with potential utility from a medical and dietary perspective [105].

## 5. Conclusions

Considering recent discoveries, *Cannabis sativa* L. hemp seeds have captured the attention of researchers and the food industry, evolving from being a byproduct of the industry to a primary source of nutrients and bioactive compounds. This transition has been catalyzed by their distinct physical, chemical, technological, and functional profile, opening significant opportunities for innovation in the food industry.

Recent research has highlighted two significant aspects of hemp seeds. Firstly, the proteins in hemp seeds do not cause food allergies and possess remarkable nutritional value, comparable to that of soy protein or even egg white. Secondly, the oil obtained from these seeds is a source of polyunsaturated fatty acids and bioactive compounds, bringing significant health benefits. The techno-functionality of hemp seeds paves the way for developing innovative foods beyond basic nutritional requirements.

However, to fully realize their potential, further studies are necessary to characterize and enhance the functional properties and flavor profile of hemp proteins. Additionally, the importance of managing CBD in hemp-derived products is emphasized. The recommended daily intake and neuroprotective effects of CBD raise discussions on therapeutic possibilities and applications in natural medicine, opening another direction for research and development.

*Cannabis sativa* L. hemp seeds represent a valuable resource for the food industry, offering versatility and nutritional quality. However, continuous research is necessary to understand all seed components’ functionality and ensure their use in safe, innovative products.

## Figures and Tables

**Figure 1 molecules-29-02097-f001:**
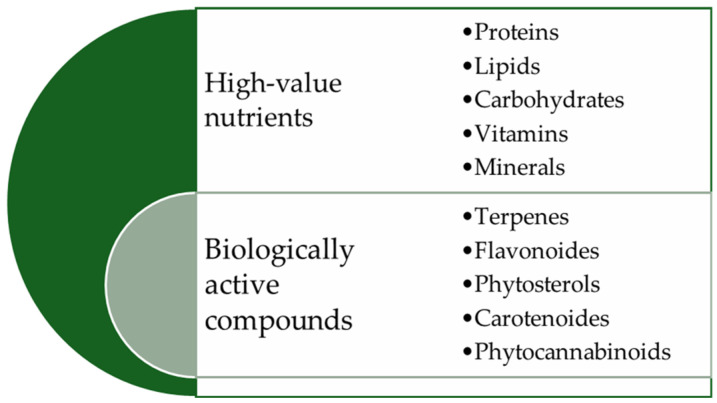
Various nutrients and bioactive compounds from hemp seeds (*Cannabis sativa* L.).

**Figure 2 molecules-29-02097-f002:**
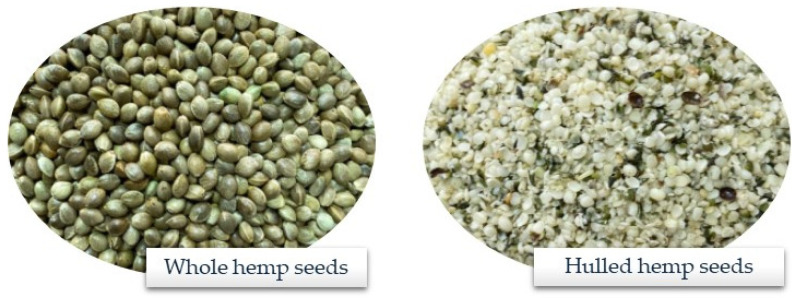
Hemp seeds (*Cannabis sativa* L.).

**Table 1 molecules-29-02097-t001:** Physical properties of hemp seeds (*Cannabis sativa* L.) [13,14,29].

Physical Properties	Description
Size and Shape	Small seeds, oval or elongated in shape, slightly asymmetric and flattened, with a size approximately 3–4 mm in length
Color	Hemp seeds are brown, light brown, or pale gray; the color may vary depending on the plant variety and the chosen method of seed harvesting.
Texture	The edible part is the interior, which has a creamy and soft texture, while the exterior part, the shell, is hard, smooth, and thin.
Markings	Hemp seeds naturally have small marks or darker stripes on the outer surface, which may vary in intensity.
Shine	Hemp seeds are glossy, with a slight shine that gives them a polished appearance.
Smell	Hemp seeds have a pleasant nutty smell
Taste	Hemp seeds have a smooth, slightly sweet, nutty taste
Hardness	The outer shell of hemp seeds is hard and requires cracking or processing to access the soft, edible interior.

**Table 4 molecules-29-02097-t004:** Terpenes identified in hemp seeds oil (*Cannabis sativa* L.) [21,71,72].

Volatile Compounds	Aroma
α-pinene	terpenic, sweet pine, woody, earthy
β-pinene	fresh, dry woody, pine
β-myrcene	pine, resin, turpentine
D-limonene	lemon, orange
β-caryophyllene	wood, spice
trans-β-ocimene	citrus, herb, flower
α-terpinolene	fresh woody, sweet pine, citrus
α-humulene	wood, beer-like
L-limonene	citrus, herbal, mint
sabinene	citrus, fresh
α-phellandrene	citrus, black pepper
linalool	oily, fruity, green

**Table 5 molecules-29-02097-t005:** Selected health functions of hemp seed’s nutrients.

Nutrients	Benefits to Human Health	References
Bioactive peptides	Antioxidant activity	[2]
	Antihypertensive	[14]
	Antimicrobial effect	[29]
	Hypocholesterolemic effect	[48]
	Hypoglycemic activity	[57]
Arginine	Vasodilatory properties	[14]
Polyunsaturated omega-3 and omega-6 fatty acids	Antioxidant activity	[19]
	Reduce arrhythmias and heart disease	[54]
	Support the health of blood vessels	
	Support nervous system	
Dietary fibers	Positive effects on the digestive tract	[48,49]
	Hypocholesterolemic effects	[14]
	Hypoglycemic activity	
Terpenes	Protect against oxidative stress	[19,73]
	Improve well-being (reduce physical pain and tension, balance mood and appetite control, improve cognitive functions, and support sleep and anxiety management)	[2,74]
	Anti-inflammatory and anticatabolic effects	[22]
	Enhance insulin secretion, sensitivity, and glucose uptake	[75,76]
	Lowers triglyceride and cholesterol levels	
	Maintain lipid homeostasis	
	Hypolipidemic activity	
Flavonoids	Anticancer properties	[20]
	Anti-neuroinflammatory effect	
	Antioxidant activity	[19]
	Anti-inflammatory activities	[22]
	Maintain cardiovascular health	[57,77]
Phytosterols	Hypocholesterolemic effect	[48,49]
	Anti-inflammatory effect	[54]
	Maintain cardiovascular health	[22,57]
Carotenoids	Antioxidant activity	[13,19]
	Anti-neurodegenerative effect	[34]
	Improve bone health	[2]
Cannabidiol	Pain management	[79,86]
	Positive effects on the digestive tract	
	Anxiolytic, antidepressant	
Vitamins and minerals	Antioxidant activity	[19,50]
	Hypoglycemic activity	[67]
	Improve bone health and immune system	
	Improve nervous system function	[2,67,68]

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
