# Peer review of "Hemp Seeds (Cannabis sativa L.) as a Valuable Source of Natural Ingredients for Functional Foods—A Review"

_molecules, 2024, doi:10.3390/molecules29092097_

Round 1
Reviewer 1 Report
Comments and Suggestions for Authors
The review paper described the multivalue of Hemp seeds. The contents were appropriate and pratical. But I cannot distinguish the present review and other Hemp review. Please add some highlights which are different from other Hemp review on Title or subtitles. For example, you may focus on industry/biotechnology aspect (page 13-14)? I suggest authors to add some figures of this plant (Table 1) which can increase and enrich the visibility of the review. In addition, the function of Hemp seeds should be shown in a Table.
Other issues: Figure 1 was too small. Figure 2 was not clear and a little bit small.
Author Response
The authors acknowledge the reviewer’s comments and suggestions that significantly improved the clarity and quality of the manuscript. Therefore, the authors thank the reviewer for the valuable comments and for pointing out the mistakes. The entire manuscript was carefully revised, and the corrections are highlighted in blue in the revised version.

Reviewer 2 Report
Comments and Suggestions for Authors
The present review paper is focused on hemp seeds (Cannabis sativa L.) as a valuable source of natural ingredients for functional foods. Hemp seeds do present a remarkable content of quality proteins, namely edestin and albumin, two types of proteins that contribute to exceptional nutritional value. Besides, hemp seeds are also rich in healthy lipids with a high content of polyunsaturated fatty acids, especially omega-6 and omega-3, along with some vitamins viz. vitamins E, D, and A. The review examines the scientific literature regarding hemp seeds' physicochemical and nutritional characteristics beneficial for its use in functional foods.
The review is surely interesting considering that the global Cannabis sativa market, including essential oils, foods, personal-care products, and medical formulations has gained much attention over the last years due to the favorable regulatory framework.
English is satistactory.
Lines 61-64. In reference to previous studies on the sustainability of industrial hemp, along with the extraction methods of bioactive compounds, the characterization of hemp oil, the nutritional and chemical composition of hemp seeds, some key references must be added. For instance, please include the very interesting work by Micalizzi et al. (J. Chromatogr. A 1637, 2021, 461864) who reported the analytical methodologies for cannabinoids and terpenes characterization. In particular, conventional and innovative extraction protocols, and chromatographic separations, such as GC and HPLC, were reviewed highlighting their respective advantages and drawbacks.
Lines 78-83. The authors in their review considered many scientific databases and excluded all works which did not not provide updated data or align with the objectives of the article. To this regard, the choice of the listed databases should be emphasized. For instance why Scopus was not included?
Lines 84-86. All biological compounds must be listed.
Section 3. Nutritional characterization of hemp seeds (Cannabis sativa L.). The text should reflect the classification of the various nutrients and bioactive compounds from hemp seeds as reported in Figure 1. As a consequence, I would include two main subsections: “High-value compounds” and “Biologically active compounds”, both reporting the occurring specific nutrients.
Comments on the Quality of English LanguageMinor revision.
Author Response

(The authors gave the same response as above.)

Round 2
Reviewer 1 Report
Comments and Suggestions for Authors
The manuscript has been improved by authors, however, I still suggest them to change the colour or presentation form of the figure 3.
Author Response
Dear Reviewer,
The authors would like to thank you for the suggestion made. To improve the quality of the manuscript, we decided to remove Figure 3. Please find enclosed a revised version of our manuscript.
Yours sincerely,
Daniela Istrati
